



# Advancing Urban Heat Vulnerability Assessment through SAR-Derived Vegetation and Soil Moisture Indicators: A Spatial Modelling Framework for Dhaka, Bangladesh

Aishia Fyruz Aishi[1], Md. Raffayatul Islam Khan[1], Md. Ariful Islam[2]

[1]Department of Meteorology, University of Dhaka, Dhaka, 1000, Bangladesh
[2]Advanced Chemical Industries Limited, Dhaka, 1212, Bangladesh

*Correspondence to*: Aishia Fyruz Aishi (aishia.fyruz@du.ac.bd)

**Abstract.** Urban heatwaves are intensifying due to climate change, posing significant risks to public health and infrastructure in densely populated cities. This study develops a spatially explicit framework to assess urban heat
vulnerability in the Dhaka Metropolitan Area (DMA), Bangladesh, by integrating vegetation and soil moisture indicators derived from Synthetic Aperture Radar (SAR). Sentinel-1 imagery was used to compute the Radar Vegetation Index (RVI) and estimate surface soil moisture (SSM) through empirical modelling, combining a modified Water Cloud Model (mWCM) with regression calibration against SMAP data. MODIS-derived Land Surface Temperature (LST) was used to characterize thermal variation. A Geographically Weighted Regression (GWR) model, supported by Principal Component Analysis
(PCA), quantified local relationships between LST, RVI, and SSM. Spatial autocorrelation analysis using Moran's I confirmed clustering in both thermal and environmental variables. Results show that areas with higher vegetation and soil moisture correspond to lower LST, highlighting their cooling effects. The model achieved strong performance (R² = 0.8835; RMSE = 0.6126; MAE = 0.4753), demonstrating its robustness and applicability in data-scarce contexts. A Heat Vulnerability Index (HVI) was constructed to spatially map susceptibility to extreme heat. This SAR-based approach
supports targeted urban heat adaptation strategies through spatially informed planning.

## 1 Introduction

Urban areas worldwide are experiencing intensified warming due to the combined effects of global climate change and rapid urbanization (Cheval et al., 2024). This phenomenon, known as the Urban Heat Island (UHI) effect, results in higher temperatures in urban regions compared to their rural surroundings, exacerbating heat-related health risks and energy
demands (Diem et al., 2024). Recent studies have highlighted the increasing frequency and intensity of heatwaves in cities, emphasizing the need for comprehensive assessments of urban heat vulnerability (Deng et al., 2024; Liu et al., 2025). Dhaka, the capital of Bangladesh, is one of the fastest-growing megacities in the world, facing significant challenges related to urban heat (S. Islam et al., 2024). The city's dense population, rapid urbanization, and limited green spaces contribute to



pronounced UHI effects (Shanto et al., 2025). Recent research indicates that Dhaka experiences higher temperatures
compared to its rural surroundings, with implications for public health and urban liveability (Abrar et al., 2022).

Several recent studies have examined the urban heat island (UHI) effect in Dhaka, highlighting its spatial heterogeneity and
underlying drivers. Dewan et al. (2021) found that, until 2019, the mean annual daytime surface UHI intensity (SUHII) in
Dhaka was 2.88 °C, with surface imperviousness and vegetation loss acting as key determinants. Tabassum et al. (2024)
employed the Weather Research and Forecasting (WRF) model to simulate thermal and wind environments in Dhaka,
demonstrating that anthropogenic heat and urban morphology amplify UHI effects, particularly at night. Recent geostatistical
approaches, such as hotspot analysis of remotely sensed land surface temperature (LST) data, have shown substantial
increases (93.73%) in thermal hotspots across Dhaka, driven by land use changes from 1991 to 2015 (Hussain et al., 2023).
Another study by Abrar et al. (2022) utilized a heat vulnerability index (HVI) based on 26 demographic and environmental
indicators, analysed through principal component analysis (PCA), to identify high-risk zones at the sub-city level. Shanto et
al. (2025) presented a 23-year trend analysis of LST in Dhaka, highlighting an alarming rise of up to 1.98 °C during summer
seasons, largely attributable to increased impervious surfaces and diminishing vegetation.

Radar remote sensing, particularly using SAR data from Sentinel-1, has gained prominence for its ability to retrieve key
biophysical variables even under cloudy and low-light conditions (Aishi et al., 2023). Among the SAR-based indices, the
Radar Vegetation Index (RVI) has proven effective for assessing vegetation dynamics and surface conditions in urban and
peri-urban environments (Oh & Kim, 2014). RVI exploits the dual-polarization capacity of Sentinel-1 (VV and VH) to
delineate vegetation structure and moisture variability, offering complementary insights to optical indices such as NDVI
(Holtgrave et al., 2020). This index is particularly useful in densely built environments where vegetative cover is spatially
heterogeneous and temporal changes are rapid (Hu et al., 2024). Sentinel-1 data have also been applied in the retrieval of soil
moisture through various approaches, including statistical models, physical scattering models, and machine learning
frameworks. One robust example is the High Spatial Resolution Soil Moisture Estimation Framework (HSRSMEF), which
integrated Sentinel-1 SAR and Sentinel-2 optical data in Google Earth Engine (GEE) to map soil moisture with high
accuracy (Guo et al., 2023). Similarly, synergistic use of multi-source Sentinel data, as demonstrated by Madelon et al.
(2023), provided 1-km resolution surface soil moisture estimates for operational hydrological modelling and climate risk
mapping. Furthermore, recent advances in knowledge-guided deep learning have been employed to estimate field-scale soil
moisture from Sentinel-1 SAR data, thereby overcoming some limitations of empirical backscatter–moisture relationships
(Yu et al., 2025). Kim & van Zyl (2009) first included vegetation canopy water content retrieval using time-series dual-
polarized SAR data, paving the way for integrated vegetation-soil moisture modelling.

Despite the significant progress made in utilizing SAR data for environmental monitoring, several gaps persist, particularly
in the context of urban heat vulnerability assessments. Numerous studies have explored UHI and associated vulnerabilities in
Dhaka; most have relied exclusively on optical imagery and conventional LST retrieval techniques using Landsat or MODIS
platforms (Dewan et al., 2021). These methods often suffer from data gaps due to persistent cloud cover during the monsoon
season. Although RVI provides a promising radar-based alternative, its application in Dhaka's urban core remains



underexplored. Moreover, while international studies like Guo et al. (2023) and Madelon et al. (2023) have developed sophisticated frameworks for soil moisture retrieval, these are rarely integrated into heat vulnerability models. There is a lack

of standardized methodology to incorporate SAR-derived soil moisture and vegetation indices simultaneously within a spatially explicit urban heat vulnerability framework. Most existing models are either LULC-based or rely heavily on temperature proxies, often neglecting subsurface hydrological influences that radar can uniquely capture (Abrar et al., 2022). On the methodological front, spatial statistical techniques such as GWR, Moran's I, and PCA are underutilized in studies using SAR-derived variables. These tools offer powerful means to explore spatial heterogeneity, autocorrelation, and multi-

dimensional vulnerability factors.

In response to the limitations identified above, this study proposes an innovative framework for assessing urban heat vulnerability in Dhaka by incorporating radar-based vegetation and soil moisture indices and robust spatial statistical techniques. One of the key novelties lies in the operationalization of Sentinel-1 radar data for soil moisture retrieval using WCM and SMAP in the absence of in situ data. To account for spatial variation in the relationships between variables, the

study applies Geographically Weighted Regression (GWR), allowing for localized model interpretation. Further, the study employs Local Moran's I to detect spatial clusters of vulnerability, offering a statistically robust basis for identifying high-risk zones. By embedding physically meaningful and cloud-robust radar-derived variables into a geostatistical modelling framework, this study not only enhances the methodological rigor of UHI vulnerability assessment in Dhaka but also contributes a replicable, cloud-resilient framework suitable for other data-constrained, rapidly urbanizing cities in the Global

South. Its innovations respond directly to data availability challenges and analytical gaps in monsoon-prone cities and offer a novel and operationally feasible approach to inform climate-resilient urban planning.

## 2 Methodology

### 2.1 Study area

Dhaka, the capital of Bangladesh, is situated in the Ganges Delta (23°42′N, 90°22′E) and spans approximately 306.38 km².

The Greater Dhaka Area, home to an estimated population of nearly 18 million, is experiencing one of the highest urban growth rates in the world, largely driven by rapid industrial development and continuous rural-to-urban migration (Huq & Alam, 2003; World Bank, 2023). This study focuses on the Dhaka Metropolitan Area (DMA) due to its acute urban heat risks and status as a representative tropical megacity. The DMA's high population density, industrial intensity, and impervious surfaces exacerbate its UHI effect, posing critical environmental challenges (Islam et al., 2024).

Climatically, the DMA experiences a tropical monsoon climate with pronounced UHI effects. Between 1988 and 2018, summer LST rose by up to 3.76°C (March-May), linked to urbanization and vegetation loss (Begum et al., 2021). Rapid land-use changes transformed the DMA's landscape, with built-up areas expanding from 30% (1991) to >90% (2019) and average LST increasing by 3-5°C over 28 years (Faisal et al., 2022). Between 1993 and 2023, DMA lost 139.17 km² of water bodies and vegetation to urban expansion, and retained the highest amount of LST hotspot zones due to highly concentrated



urban areas (Miah et al., 2024; M. N. Rahman et al., 2022). Long-term analysis (1981-2015) reveals increasing annual maximum temperatures (0.017°C/year) and declining rainfall (-16.11 mm/year) across Dhaka Division, with urban cores experiencing sharper thermal anomalies (Khatun et al., 2017). The selection of the DMA for this analysis was based on several interrelated criteria, including its high urban density, pronounced vulnerability to extreme heat, climatic diversity, and data availability. The compact urban form of the DMA exacerbates heat accumulation and slows nighttime cooling,

resulting in persistent thermal stress, particularly in densely populated neighbourhoods (Oke, 1982; Tan et al., 2010).

**2.2 Data collection and preprocessing**

A suite of multi-source geospatial datasets was collected and processed to analyse vegetation dynamics, LST, and soil moisture patterns across the DMA for the 2016-2022 period. These datasets were selected based on spatial and temporal coverage, resolution, and suitability for urban climate analysis. All preprocessing, filtering, and harmonization tasks were

performed using Google Earth Engine (GEE), and datasets were reprojected to WGS84 and resampled to ensure consistency in spatial alignment.

**Table 1: Description of the datasets used in the study.**

| Data Type | Source | Date Range | Resolution |
| --- | --- | --- | --- |
| Sentinel-1 SAR | ESA / GEE | 2016-2022 | 10 m |
| MODIS LST (gap-filled) | GEE | 2016-2020 | 1 km |
| MODIS LST | NASA | 2021-2022 | 1 km |
| SMAP Soil Moisture | NASA / GEE | 2016-2022 | 9 km |
| Heatwave Data | BMD & NOAA | 2016-2024 | N/A |

Daily maximum temperature data were collected from the Bangladesh Meteorological Department (BMD) and supplemented with NOAA reanalysis products to ensure continuity and regional consistency. A heatwave event was defined, following

BMD criteria, as any day with a maximum temperature (Tmax) exceeding 36 °C (Rashid et al., 2024). Heatwave dates were identified using processed daily Tmax data, and events were categorized by duration and magnitude using Python in Jupyter Notebook. Annual frequency and seasonal distributions were evaluated to assess temporal patterns and shifts in extreme heat events. These identified heatwave periods were used to filter and align other datasets for targeted analysis.

Sentinel-1 C-band Synthetic Aperture Radar (SAR) data, acquired in the Interferometric Wide (IW) swath mode, were used

to derive vegetation and moisture-sensitive backscatter signals. Dual-polarization bands (VV and VH) were utilized, with VV polarization emphasized due to its sensitivity to soil moisture, while VH polarization provided complementary information on vegetation structure and volume scattering (Baghdadi et al., 2017; Torres et al., 2012). SAR data were calibrated and filtered to suppress speckle noise and perform further analysis (Filipponi, 2019).

Land Surface Temperature (LST) data were obtained from the MODIS Terra and Aqua satellites via the MOD11A1 and

MYD11A1 products. For the 2016-2020 period, significant cloud-induced data gaps were addressed using a spatiotemporal



interpolation technique based on splines and the inverse distance weighting method (Zhang et al., 2020). This gap-filling process enabled the generation of near-continuous daily LST records at 1 km spatial resolution. For the 2021–2022 period, unfilled MODIS LST data from NASA's LP DAAC archive were used directly, and quality assurance flags were applied to filter out low-quality observations.

Surface soil moisture data were retrieved from NASA's Soil Moisture Active Passive (SMAP) mission using Level-3 passive microwave products available via Google Earth Engine (GEE). These daily data, with a nominal resolution of 9 km, were filtered to exclude retrievals affected by Radio Frequency Interference (RFI) or dense vegetation, based on the quality metrics outlined by (Chan et al., 2018). Although the spatial resolution was coarser than that of other datasets, SMAP was retained due to its proven reliability in detecting surface moisture variability in areas lacking in situ measurements.

This study focused on the period from 2016 to 2022, constrained by the availability of key remote sensing datasets - specifically MODIS LST and SMAP soil moisture products, which were only accessible through the end of 2022. The selection of this timeframe ensured consistent temporal coverage across all variables of interest, enabling coherent multi-sensor analysis.

## 2.3 RVI and SSM estimation from Sentinel-1 SAR

RVI and surface soil moisture (SSM) were retrieved from Sentinel-1 SAR data to characterize vegetation cover and soil moisture conditions across the Dhaka Metropolitan Area. RVI is a quantitative index derived from polarized SAR backscatter, offering a cloud- and light-independent approach to assess vegetation density and structural complexity (Hu et al., 2024). This makes it particularly useful in urban and tropical environments, where persistent cloud cover limits optical observations (Holtgrave et al., 2020; Kim & van Zyl, 2009). In this study, RVI was calculated using the dual-polarization

backscatter coefficients of the Sentinel-1 C-band SAR, following the simplified formulation:

$$RVI = \frac{4 \cdot \sigma_{VH}^0}{\sigma_{VV}^0 + \sigma_{VH}^0}$$

Where $\sigma_{VV}^0$ and $\sigma_{VH}^0$ represent the backscattering coefficients for vertical-vertical and vertical-horizontal polarizations, respectively. High RVI values correspond to dense or complex vegetation structures, while lower values indicate bare surfaces or impervious urban areas.

The Water Cloud Model (WCM), originally developed by Attema & Ulaby (1978), describes total radar backscatter as a combination of vegetation and soil contributions:

$$\sigma^0 = \sigma_{veg}^0 + T^2 \cdot \sigma_{soil}^0$$

Where $\sigma_{veg}^0$, $\sigma_{soil}^0$ and $T^2$ are vegetation and soil backscatter components and two-way attenuation through the vegetation canopy, respectively. Later refinements introduced vegetation descriptors (e.g., LAI, NDVI) and empirical coefficients to

better capture the soil-vegetation interaction (Dubois et al., 1995; Khellouk et al., 2021; Kumar et al., 2015). More recently, Singh et al. (2023) proposed a modified WCM (mWCM) that incorporates first-order vegetation-soil interaction:





$$\sigma_{total}^0 = \alpha \cdot \sigma_{veg}^0 + \beta \cdot \sigma_{soil}^0 + \gamma \cdot \sigma_{int}^0 + \varepsilon$$

Where $\alpha$, $\beta$, $\gamma$ are empirically derived scaling constants and $\sigma_{int}^0$ accounts for interaction effects. This model has shown improved accuracy in estimating both soil moisture and vegetation parameters. Building upon the principles of the original

and mWCM, a simplified form was adopted in this study to estimate SSM in GEE, integrating Sentinel-1 VV-polarized backscatter with RVI. Initial preprocessing involved calibrating the SAR data and applying the Lee speckle filter. A simple linear regression model was then constructed using temporally overlapping SMAP SSM data. To reduce scale differences and improve the interpretability of the regression, both Sentinel-1 VV backscatter ($\sigma^0$) and SMAP SSM values were then normalized across all images using a min-max scaling function:


$$\sigma_{VV(norm)}^0 = \frac{\sigma^0 - \min(\sigma^0)}{\max(\sigma^0) - \min(\sigma^0)}$$

$$SSM_{SMAP(norm)} = \frac{SSM_{SMAP} - SSM_{min}}{SSM_{max} - SSM_{min}}$$

This step ensures that both variables lie between 0 and 1, helping stabilize the regression fit. Then, the linear regression was performed between $\sigma_{VV(norm)}^0$ and $SSM_{SMAP(norm)}$, the resulting calibration equation is:

$$SSM_{SMAP(norm)} = a + b \cdot \sigma_{VV(norm)}^0$$

Where a and b are the intercept and slope of the linear fit, respectively, both derived from the least-squares linear regression on the training dataset. To enhance the accuracy of the retrieval, vegetation effects were incorporated into the model using RVI, resulting in a modified version of the WCM:

$$SSM = a + b \cdot (\sigma_{VV(norm)}^0 + RVI)$$

Here, $\sigma_{VV(norm)}^0$ serves as a proxy for soil moisture sensitivity, RVI approximates vegetation contribution, and the additive

structure reflects combined first-order contributions of both vegetation and soil to the total backscatter. This simplified model was chosen for operational feasibility, particularly over large and heterogeneous urban regions with limited in situ calibration data. The linear form provides a practical trade-off between physical interpretation and statistical performance, while still reflecting the dual contributions outlined in more complex WCM formulations.

**2.4 Spatiotemporal analysis**

In this study, GWR has been applied to explore the spatially varying relationships among key environmental indicators - RVI, SSM, and LST - within the DMA. Firstly, the RVI, SSM, and LST datasets were resampled to a common spatial resolution of 1 km during export from GEE, ensuring spatial alignment across variables. For each variable, multi-year temporal means were computed per pixel over the study period, resulting in a single representative raster layer. This averaging approach helped reduce noise and inter-annual variability while retaining core spatial patterns. Then all the

variables were min-max normalized to a [0,1] range to ensure scale comparability among input features. To assess multicollinearity among predictors, Pearson correlation coefficients were computed, and a Variance Inflation Factor (VIF)





analysis was conducted. VIF was calculated for the standardized variables to evaluate redundancy (O'brien, 2007). To reduce multicollinearity and condense information across the three variables, PCA - a widely accepted method for dimensionality reduction and feature importance extraction - was performed on the normalized rasters (Jollife & Cadima, 2016). Subsequently, GWR was applied to explore the spatially varying relationships between LST and the two principal components derived from RVI and SSM (Brunsdon et al., 1996). The first principal component (PC1), which captured the majority of variance across predictors, was retained as a composite environmental stressor. This ensured interpretability and numerical stability in the GWR model (Lu et al., 2014). The simplified form of the GWR model used is:

$$LST = \beta_0 + \beta_1 \cdot PC1 + \varepsilon$$

Where PC1 is the principal component score summarizing RVI and SSM at a specific location, $\beta_0$ and $\beta_1$ are location-specific intercept and slope terms, and $\varepsilon$ is the model residual. An adaptive kernel was used, which adjusts the bandwidth depending on local point density, ensuring a better fit in both dense and sparse areas. The optimal bandwidth was selected using corrected Akaike Information Criterion (AICc) to balance model complexity with fit. GWR allows for the estimation of local regression coefficients, enabling analysis of non-stationary spatial interactions between variables across the urban landscape. To evaluate the predictive performance of the GWR model, several statistical metrics were utilized, including residuals, Mean Absolute Error (MAE), Root Mean Squared Error (RMSE), and the coefficient of determination ($R^2$). Additionally, spatial residual maps were generated to highlight regions where the GWR performed poorly, helping to identify zones where explanatory variables were insufficient or other unmodeled factors might be at play.

Although multi-temporal mean rasters were used for the main modeling components, trends were also examined using the available date-wise composite data layers to investigate both spatial and temporal dynamics. GWR models were also fitted separately for each composite dataset. This allowed for the analysis of how localized relationships between vegetation, soil moisture, and land surface temperature evolved across the DMA. Comparing these outputs enabled the identification of persistent versus shifting spatial patterns in urban climate interactions. Additionally, Moran's I was calculated separately for each date-wise composite raster of LST, RVI, and SSM to evaluate spatial autocorrelation (He et al., 2019). This analysis quantified spatial clustering patterns for each observation date, revealing whether high or low values were spatially aggregated across the metropolitan area.

## 2.5 Heat Vulnerability Index (HVI) calculation

To derive objective and data-driven weights for the construction of the HVI, PCA was employed on the standardized dataset. These weights were then used to calculate the HVI using a linear combination:

$$HVI = w_1 \cdot LST + w_2 \cdot SSM + w_3 \cdot RVI$$

Where $w_1$, $w_2$, and $w_3$ are the PCA-derived weights. This method avoided subjective assumptions and ensured that the relative importance of each variable in the HVI reflected empirical variance contributions. The resulting HVI raster was classified into three vulnerability categories - Low, Moderate, and High - based on quantile breaks to ensure balanced spatial



distribution among classes. This approach facilitated the identification of urban areas with disproportionately high
environmental heat stress, supporting targeted intervention and adaptation strategies in urban planning.

## 3 Results and discussion

### 3.1 Extreme heat events identification

The analysis conducted for the period 2016-2024 reveals a notable increase in both the duration and frequency of heatwave
events in the Dhaka Metropolitan Area (DMA). Particularly, the later years of the study showed a marked rise in multi-day
heat events. Seasonal conditions - such as peak solar radiation, reduced precipitation, and stagnant atmospheric circulation -
contribute to conducive environments for extreme heat accumulation. As illustrated in Fig. 2(b), substantial year-to-year
fluctuations were observed. The years 2020 and 2021 recorded the highest number of days exceeding the 36 °C threshold,
indicating particularly severe and prolonged heat events. In contrast, 2018 and 2023 experienced fewer extreme temperature
days, implying temporary moderation of heat stress. These patterns align with the findings of Prodhan et al. (2024).

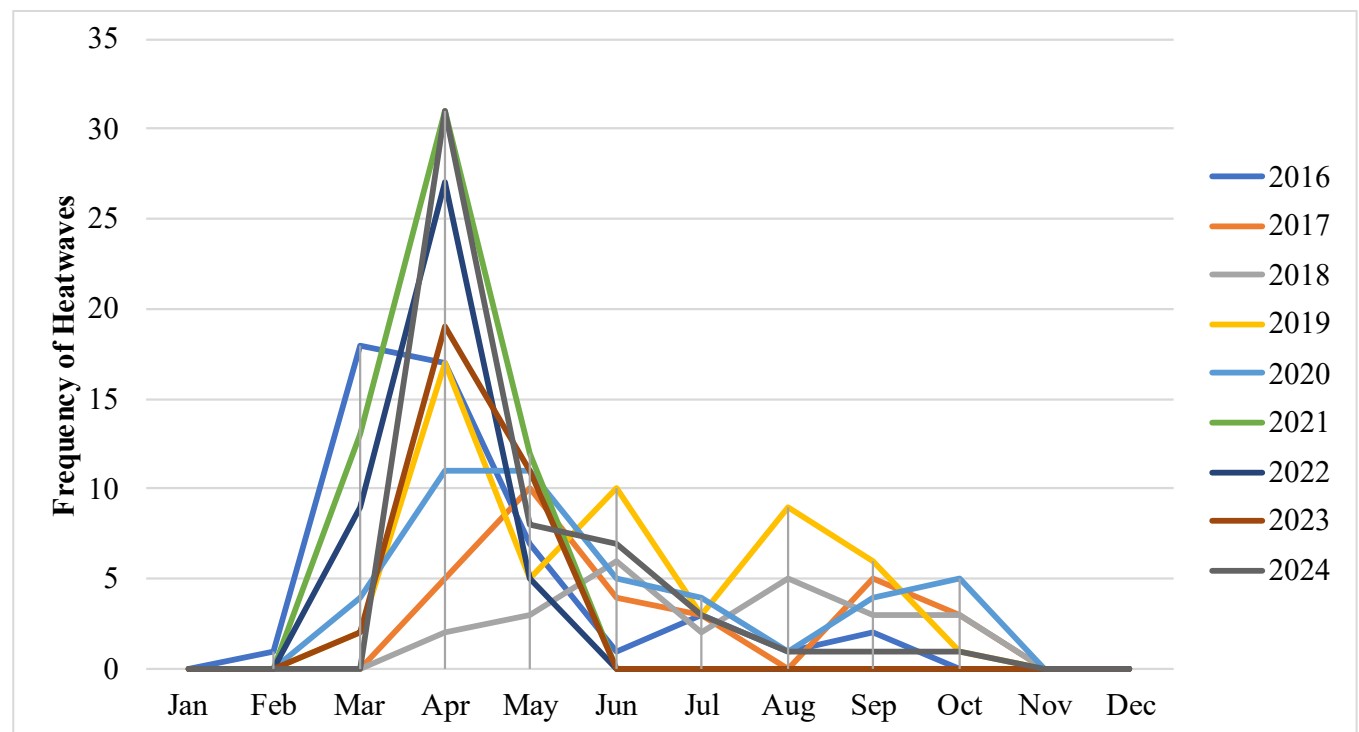


**Figure 1: Monthly distribution of the number of heatwave days across the DMA from 2016 to 2024.**

Figures 1 and 2(a) show that extreme heat days cluster heavily in March-May, with April being the peak month for heatwave
occurrence in DMA. This coincides with the pre-monsoon season, when conditions favour heat build-up before the arrival of
summer rainfall (Rahman et al., 2024). Following this period, a consistent decline in heatwave events gives way to the



monsoon season (June-October), during which cloud cover and rainfall substantially reduce surface temperatures. Conversely, November to February exhibit no heatwave days, reaffirming the cooler winter pattern typical of Bangladesh (Rashid et al., 2024; Tabassum et al., 2024). The growing prevalence of long-duration heatwaves (≥3 days) from 2020 onward reflects a heightened urban heat vulnerability, especially the occurrence of 15+ consecutive days of heatwave in 2021 and 2022 (Fig. 3). These prolonged events pose increasing challenges for public health, urban infrastructure, and heat-

risk management, especially among socio-economically vulnerable communities (Rashid et al., 2024). But over these two years, the frequency of shorter-duration heatwaves was lowest despite the rising trend post-2019.

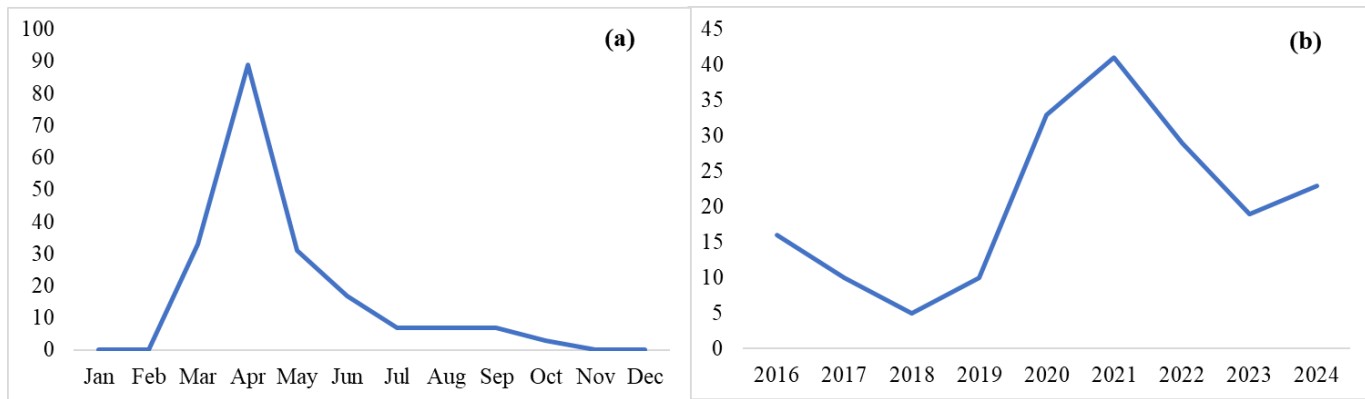

**Figure 2: (a) Cumulative monthly heatwave days in the DMA from 2016 to 2024, indicating seasonal heatwave patterns; (b) Annual total number of heatwave days recorded in the DMA.**

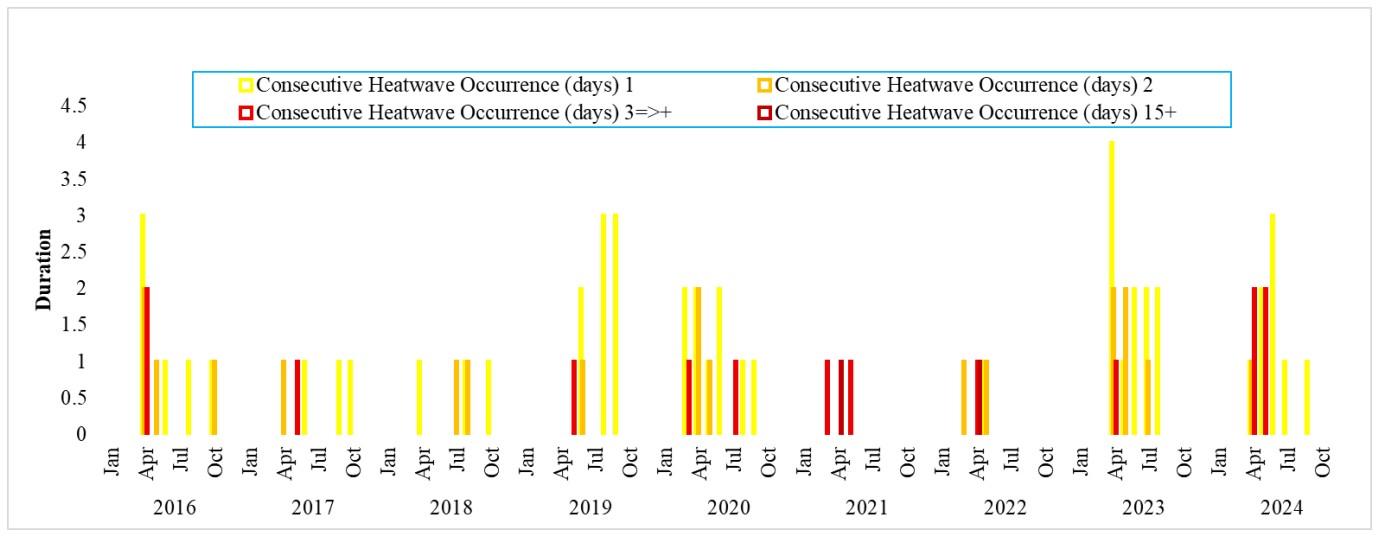


**Figure 3: Yearly breakdown of monthly heatwave days from 2016 to 2024, categorized by event duration - 1 day, 2 consecutive days, 3 or more consecutive days, and extended periods of 15+ days.**



## 3.2 Autocorrelation analyses

Figure 4 shows GEE exports of LST, SSM, and RVI maps on representative dates. These maps were then stacked date-wise

and analysed across multiple available dates to evaluate spatiotemporal environmental stressors.

**Figure 4: Maps showing LST, SSM, and RVI for three representative dates - 12 April 2017, 7 June 2020, and 17 March 2022.**



Spatial autocorrelation was assessed using Moran's I across multiple single-date composites for LST, SSM, and RVI during 2016-2022. Moran's I values close to +1 indicate strong clustering; 0 implies spatial randomness; values near −1 suggest
dispersion. LST consistently shows high positive spatial autocorrelation, with values ranging from 0.5 to 0.6 in most years, peaking at 0.6304, reflecting persistent clustering of heat in urban zones and reinforcing the UHI effect. These findings corroborate prior studies (Rizwan et al., 2008). SSM shows the strongest autocorrelation, reaching a Moran's I of 0.8797, indicating large, homogeneous patches of similar moisture conditions. Such clustering suggests stable hydrological patterns and supports vegetation growth, which can counter urban heating (Brocca et al., 2007). RVI consistently displayed the
lowest Moran's I values, typically ranging between 0.20 and 0.35. These relatively low values indicate weak to moderate positive spatial autocorrelation, suggesting that RVI is more spatially heterogeneous than SSM or LST. LST's clustering remained stable throughout the study period. RVI shows a declining trend in Moran's I from 2016 to 2020, with further instability after 2021, reflecting increased spatial fragmentation of vegetation, potentially due to urban development. SSM remained strongly clustered, but some variability across different times of the year is seen, possibly due to seasonal rainfall.
Together, these autocorrelation results suggest a dynamic interplay between urbanization, green cover loss, and heat vulnerability. The stable clustering of high LST values underscores the persistence of heat islands, while fluctuations in RVI and SSM clustering point to environmental changes affecting thermal comfort. These results reinforce the necessity of integrated urban planning focused on enhancing green infrastructure to regulate temperature extremes and improve resilience to heatwaves (Bowler et al., 2010; Gill et al., 2007).

**3.3 GWR and predictive modelling**

GWR accounts for local variation by calibrating a regression equation at each spatial location, weighted by proximity (Lu et al., 2014). In this study, GWR was employed to explore the spatially and temporally varying influence of SSM and RVI on LST across the DMA. Even though the GWR coefficients for RVI exhibited spatial and temporal variability, the coefficients are mostly positive, which aligns with conventional expectations where vegetation mitigates surface heating via
evapotranspiration and shading. On the other hand, GWR coefficients for SSM were consistently negative across most periods, reinforcing the hypothesis that higher soil moisture has a cooling effect on surface temperatures. This pattern was especially prominent in early 2021, indicating strong temporal variability in the soil moisture-LST relationship. The persistence of negative coefficients highlights the significance of soil moisture in regulating urban thermal environments.

The spatial distribution of GWR coefficients is presented through a heatmap in Fig. 5. Darker red regions indicate strong
negative coefficients, suggesting a greater influence of environmental factors (SSM, RVI) in mitigating high LST values. These are often concentrated in areas with limited vegetation or dense built-up environments. The Dhaka city center, lying in the deepest red zone, is indicative of the UHI effect. Conversely, green-shaded regions exhibit positive coefficients, indicating lower thermal stress, likely due to better vegetative cover or more reflective surfaces (Yang et al., 2005). But the peri-urban zones mostly lie where PC1 shows near 0 values, which covers the majority of the DMA. The central and eastern




parts of the DMA show steep spatial gradients in coefficient values, revealing heterogeneous urban development patterns and uneven green infrastructure.

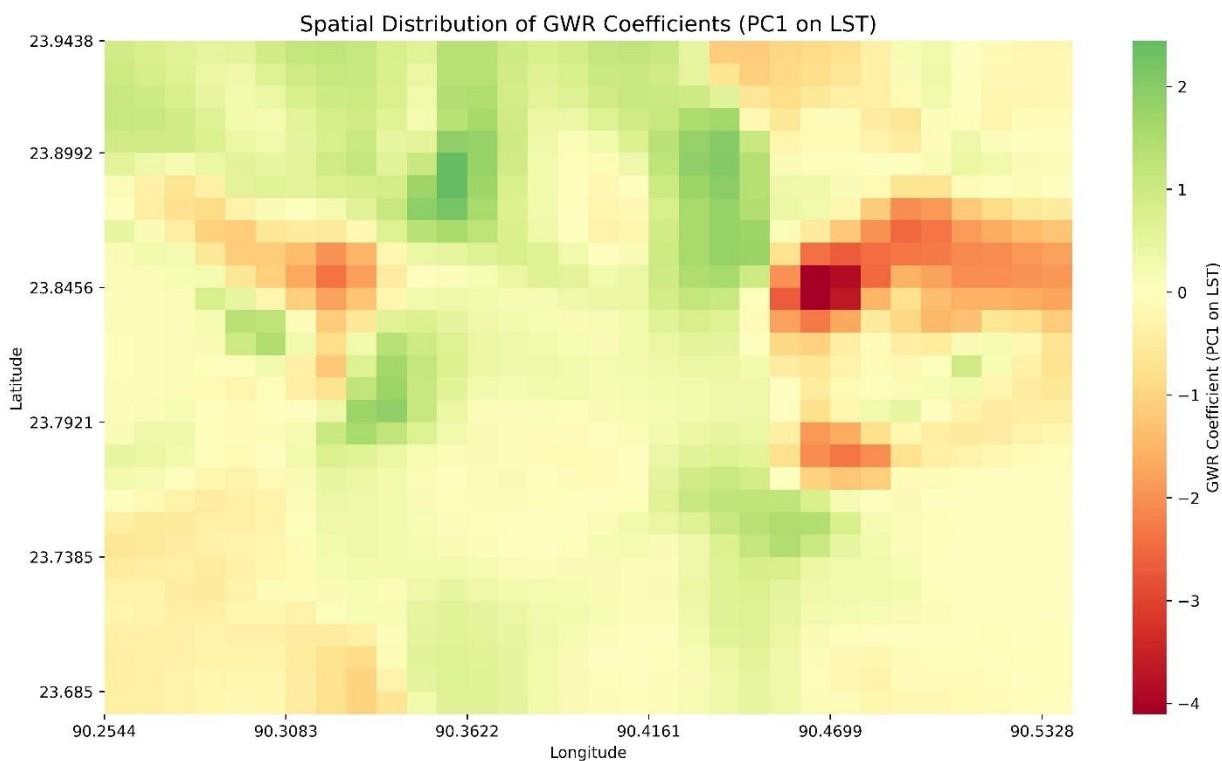

**Figure 5: Spatial distribution of GWR coefficients for the principal component of LST across the DMA, indicating the localized influence of SSM and RVI.**

**3.4 Model accuracy assessment**

The residual analysis from the GWR model offers critical insights into its predictive reliability. Mapping the spatial distribution of residuals (Fig. 6) identifies zones where the model either overestimates or underestimates LST. The blue zones show positive residuals, indicating underestimation of actual LST. This discrepancy may be attributed to localized heat sources or microclimatic effects not captured by the predictor variables (Fotheringham et al., 2009). On the other hand, red

zones indicate negative residuals, suggesting the model over-predicts LST. These areas may benefit from unaccounted cooling factors that were not modeled (Brunsdon et al., 1996). Areas with near-zero residuals suggest good agreement between observed and predicted LST values, attesting to the model's spatial accuracy under certain conditions. Figure xx shows that most areas have near-zero residuals.



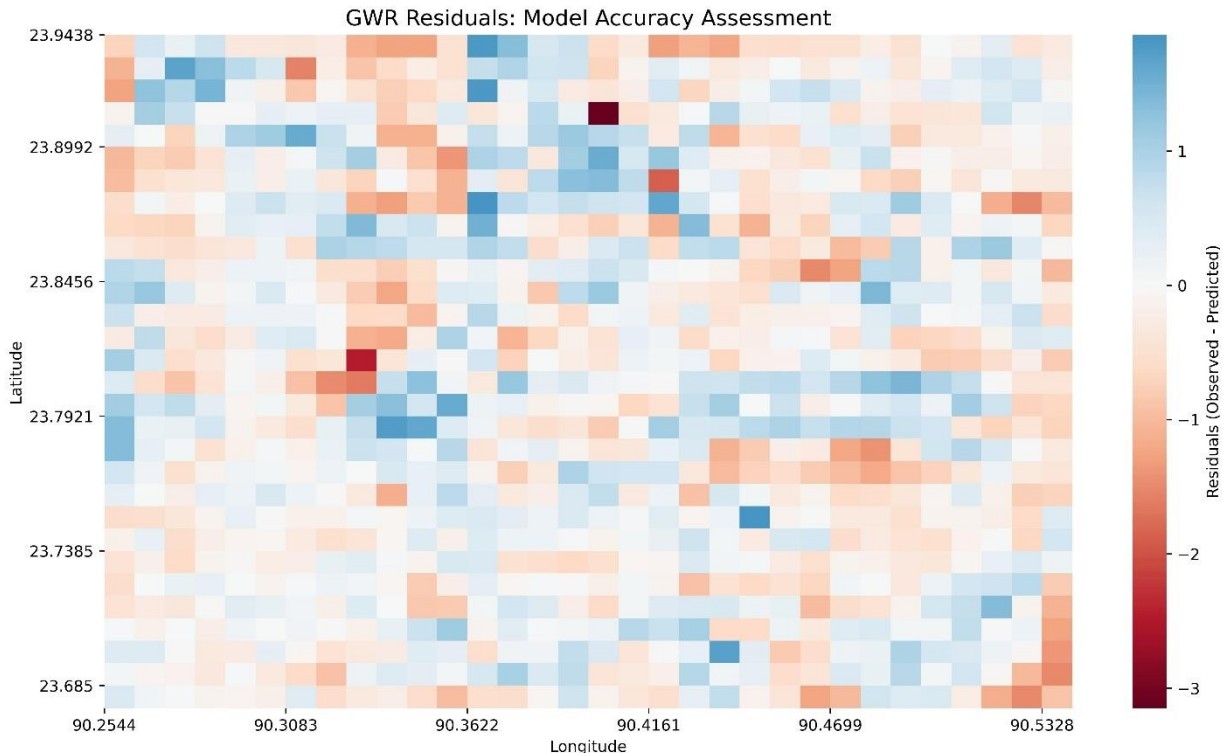

**Figure 6: Residual distribution from the GWR model across the DMA, highlighting areas of overestimation and underestimation in predicted LST values.**

The accuracy of the GWR model was evaluated using several key statistical metrics. The Mean Absolute Error (MAE) of 0.4753 indicates a low average prediction error, suggesting that the model performs well in capturing the general behavior of the data without being overly influenced by outliers (Willmott & Matsuura, 2005). The Root Mean Squared Error (RMSE) is 0.6126, reflecting high model accuracy and the model's effectiveness in minimizing larger prediction errors (Chai & Draxler, 2014). Most notably, the $R^2$ score of 0.8835 demonstrates that the model explains approximately 88.35% of the variance in LST, indicating strong predictive power and a robust fit to the spatial data (Fotheringham et al., 2009). The high $R^2$ and low error metrics indicate that the GWR model effectively captures spatial LST variations using SSM and RVI. Overall, the GWR model proves robust for spatial LST prediction in DMA.

**3.5 Heat vulnerability mapping of the DMA**

The Heat Vulnerability Index (HVI) map provides a spatially explicit classification of vulnerability across the DMA, categorizing regions into high, moderate, and low vulnerability zones (Fig. 7). This framework enables targeted planning for heat risk mitigation. Hotspots such as Uttar Khan, Khilgaon, Demra, and parts of Turag exhibit elevated HVI values. These regions are characterized by high population density, extensive built-up areas, and limited vegetative cover - conditions that intensify the UHI effect. These zones demand immediate interventions such as heat-resilient infrastructure, greening



initiatives, and heat warning systems. Areas like Uttara, Cantonment, Badda, and Kotwali fall under moderate vulnerability. These districts exhibit transitional land use with varying vegetation and built-up cover. Proactive efforts - such as improving urban greenery, promoting water-sensitive urban design, and regulating construction density - can help prevent further escalation of heat exposure (Harlan et al., 2006). Peripheral neighborhoods such as Daksahinkhan, Mohammadpur, Adabor,

and Tejgaon demonstrate low HVI scores, attributable to greater vegetation density, lower urban density, or the presence of open green areas. These areas serve as cooling buffers and should be preserved to sustain ecosystem services and thermal regulation (Gill et al., 2007). The spatial gradient from high vulnerability in central areas to lower vulnerability on the periphery underscores the impact of urban morphology on heat risk. The HVI map thus serves as an essential decision-support tool for urban policymakers. It guides spatial prioritization for heat mitigation and adaptation, enabling equitable

allocation of resources to safeguard at-risk communities and promote climate-resilient urban growth.

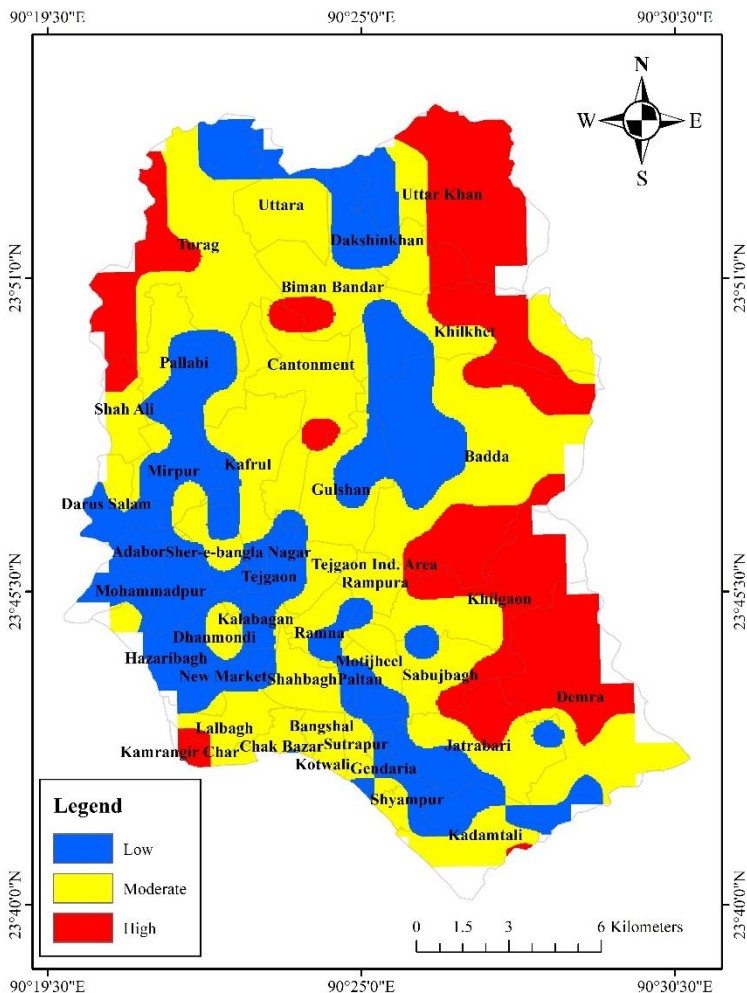

**Figure 7: HVI zonation map of the DMA, classifying regions into low, moderate, and high vulnerability to urban heat stress based on LST, SSM, and RVI.**



## 4 Conclusion

This study employed GWR to explore the spatial heterogeneity in the relationship between LST, RVI, and SSM across the DMA. By integrating radar-derived indices and spatial regression techniques, the analysis reveals critical spatial patterns that underpin the UHI phenomenon in one of South Asia's most rapidly urbanizing megacities. The GWR results underscore the localized cooling effects of soil moisture, with consistently negative coefficients across most periods, reaffirming its moderating role on urban surface temperatures. Conversely, RVI exhibited more spatially and temporally variable

relationships with LST, reflecting the complex influence of vegetation structure, density, and seasonal dynamics. The spatial distribution of GWR coefficients highlights distinct thermal gradients within the city, with higher thermal vulnerability concentrated in densely built-up central zones and relatively lower stress in vegetated or peri-urban areas. This spatial differentiation was further supported by the accuracy metrics, indicating strong model performance.

The generated HVI map offers a valuable spatial decision-making tool, delineating high-risk zones such as Khilgaon, Demra,

and Uttar Khan, where mitigation efforts should be prioritized. These findings have significant implications for climate-resilient urban planning, particularly in integrating green infrastructure, enhancing surface reflectivity, and preserving natural soil moisture regimes. As extreme heat events become more frequent under ongoing climate change, the approach and results presented here contribute to a data-driven framework for managing urban thermal risk in developing megacities. Future work should incorporate dynamic socioeconomic variables and high-resolution urban morphology data to better

capture the multidimensional nature of urban heat exposure. Nonetheless, the spatially explicit insights from this study offer a critical foundation for adaptive planning and equitable heat mitigation strategies in Dhaka and similar urban environments.

### Code and data availability

MODIS LST and Sentinel-1 data used in this study are publicly available from GEE platform. All data used and produced in this study, including heatwave records, LST, SSM and RVI, are available upon reasonable request from the corresponding

author. Much of the spatial analysis, including time-series extraction and preprocessing, was conducted using GEE. The scripts used for autocorrelation analysis and model development have been developed in Python. Code scripts developed in GEE JavaScript API and Python can be shared for academic purposes by contacting the authors directly.

### Interactive computing environment

Spatial data preprocessing and temporal compositing were primarily conducted using GEE due to its high-performance

cloud-based processing capabilities and access to multi-source Earth observation data. Statistical modelling and validation metrics was performed in Python 3.10 using mgwr, pysal, and geopandas libraries. Visualization and layout design were done using ArcGIS.

**Sample availability**

No physical or biological samples were used or generated as part of this research. All datasets are derived from publicly available remote sensing platforms, and meteorological archives.

**Author contributions**

AFA conceptualized the study and provided supervision throughout the research process. MRIK was responsible for data acquisition, preprocessing, and conducting the simulations. MAI supervised and reviewed the statistical modelling and validation procedures. AFA developed the model code. MRIK prepared the initial manuscript draft with contributions from all co-authors. All authors contributed to reviewing, editing, and approving the final version of the manuscript.

**Competing interests**

The authors declare that they have no conflict of interest.

**Acknowledgements**

The authors express their sincere gratitude to the Department of Meteorology at the University of Dhaka for providing institutional support, technical assistance, and access to computational infrastructure, where this research was conducted. Special thanks are extended to the faculty and adjunct faculty members of the department for their valuable suggestions and guidance throughout the research process. The authors also gratefully acknowledge the BMD for sharing essential temperature datasets. Additionally, this work benefited significantly from the use of open-source platforms such as GEE for data acquisition and spatial analysis.

**Financial support**

This research has been supported by the Center for Climate Change Study and Resource Utilization (CCCSRU), University of Dhaka.

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
