# Peer review of "Advancing Urban Heat Vulnerability Assessment through SAR-Derived Vegetation and Soil Moisture Indicators: A Spatial Modelling Framework for Dhaka, Bangladesh"

_EGUsphere, 2025_

## Author Comment (AC1)

**Author Responses to RC1: 'Comment on egusphere-2025-3218'**

**Reviewer Comment 1:**

*The manuscript needs major modification in order to be accepted for publication. First, a better connection between the methods used and their purpose should be explained. Including a methodological diagram could be beneficial. For example, the analysis of spatial autocorrelation and GWR and their goal are not well supported.*

**Author Response:**

We agree with this point and have revised the entire manuscript to explicitly connect each analytical step to its purpose in assessing urban heat vulnerability. We now also explain how Moran's I and LISA are used to detect spatial clustering of thermal patterns, and how GWR quantifies localized relationships between LST, vegetation, and soil moisture. A methodological workflow diagram (Figure 1) has been added to visually summarize the process from data collection to final HVI mapping.

[Figure]

**Figure 1: Methodological framework for urban heat vulnerability assessment in the Dhaka Metropolitan Area.**
* * *
**Reviewer Comment 2:**

*A clearer explanation of the temporal domain used and its purpose is needed. When should you use annual data, and when is it appropriate to use monthly or daily data? Currently, it is not clear on the document.*

**Author Response:**

As explained in Lines 110 and 175 of the preprint (Section 2.2 and 2.4), our analysis uses pixel-wise multi-year means (2016-2022) computed from daily observations, restricted to heatwave dates when all datasets were simultaneously available. This design choice focuses the analysis on extreme-heat conditions, reduces inter-annual variability, and preserves core spatial patterns relevant to heatwave impacts. The primary goal was to synchronize datasets

with differing temporal resolutions over a consistent set of dates. In general, annual averages are more suitable for assessing long-term trends, monthly data capture seasonal variability, and daily data allow for event-based analysis such as heatwaves. Our use of daily data limited to heatwave periods aligns with the study's aim of characterizing urban heat vulnerability under extreme thermal stress.
* * *
**Reviewer Comment 3:**

*The manuscript needs a study area figure with land cover types.*

**Author Response:**

A study area map (Figure 2) has been added, showing administrative boundaries, major water bodies, and detailed land cover classifications derived from ESA WorldCover 2021.

[Figure]

**Figure 2: Dhaka Metropolitan Area (DMA) divided into administrative sub-units and land cover distribution (percentage of total area per land cover class shown in pie chart). [Data source: ESA WorldCover 2021]**
* * *
**Reviewer Comment 4:**

*At the end of the introduction, it is stated that the author will use Moran's I to detect spatial*

*clusters (L75). However, in the results section, Global Moran's I is used instead. For a better identification of the clustering, you should use Moran's I as Local Indicators of Spatial Association (LISA) to achieve this.*

**Author Response:**

We have now performed a Local Moran's I (LISA) analysis for LST, RVI, and SSM. The results are presented in new Figure 7, showing high-high, low-low clusters and outliers, and discussed under section 3.2 (Autocorrelation analyses), along with the number of statistically significant locations for each variable.

[Figure]

**Figure 3: Spatial autocorrelation analysis of local indicators of spatial association (LISA) across the Dhaka Metropolitan Area (DMA). The maps show clustering patterns for Land Surface Temperature (LST), Radar Vegetation Index (RVI), and Sentinel-1 Soil Moisture (SSM).**
* * *
**Reviewer Comment 5:**

*At least a figure showing the time series of global Moran's I per variable would help the reader. Then in Fig. 4 could be used the variable and year that has the higher Moran's I.*

**Author Response:**

We have added a time series plot of Global Moran's I for LST, RVI, and SSM (new Figure 6). Additionally, we have added the variables from the date with the highest Moran's I value of LST, as suggested.

[Figure]

**Figure 4: Temporal trends of Moran's I spatial autocorrelation for Land Surface Temperature (LST), Radar Vegetation Index (RVI), and Sentinel-1 Soil Moisture (SSM) during 2016-2022 (top). The bottom panel shows the variables on 16th April 2018 when highest value was observed in LST.**
* * *
**Reviewer Comment 6:**

*The use of GWR in this study is just a downscaling for LST. But not a predictive model for LST. The absence of in-situ LST measurements explains this.*

**Author Response:**

We appreciate your insightful observation. We acknowledge the limitation posed by the lack of in-situ LST measurements, which constrains the use of GWR as a predictive modeling tool in this study. Accordingly, we have clarified throughout the revised manuscript that our application of GWR is exploratory and primarily intended to characterize local spatial relationships rather than to serve as a predictive model for LST. This clarification has been explicitly added to prevent any potential misinterpretation of the results.
* * *
**Reviewer Comment 7:**

*L15: RMSE and MAE should have units.*

**Author Response:**

Units (°C) have been added for both RMSE and MAE values in the abstract and throughout the manuscript.
* * *
**Reviewer Comment 8:**

*L120: It's unclear why two MOD11 ranges and sources are being used. Please clarify this.*

**Author Response:**

Thank you for pointing this out. To clarify, we used the MOD11A1 product, which provides daily LST data. We have revised the manuscript to explicitly state this and removed any ambiguity regarding the data sources and ranges.
* * *
**Reviewer Comment 9:**

*L175: Which resampling method was used? Bilineal? Please, specify.*

**Author Response:**

We have added in the revised manuscript that nearest neighbor was used for resampling continuous raster datasets to match spatial resolution.
* * *
**Reviewer Comment 10:**

*L220: These extreme heat events were derived from LST or just the BMD & NOAA data. Please, start with a sentence that could clarify this.*

**Author Response:**

Thank you for the suggestion. We have revised the manuscript to clearly state that extreme heat events were identified using temperature records from BMD and NOAA. The relevant section now begins: "The analysis conducted using BMD data for the period 2016-2024 reveals a notable increase in both the duration and frequency of heatwave events in the Dhaka Metropolitan Area (DMA)".
* * *
**Reviewer Comment 11:**

*L240: Please choose a proper color palette for the figure; currently the colors are very similar, and it is difficult to distinguish between them.*

**Author Response:**

The figure has been updated with high-contrast, colorblind-friendly palettes to improve visual distinction between classes:

[Figure]

**Figure 5: Yearly breakdown of monthly heatwave days from 2016 to 2024, categorized by event duration - 1 day, 2 consecutive days, 3 or more consecutive days, and extended periods of 15+ days.**
* * *
**Reviewer Comment 12:**

*L245: You do not need to indicate "GEE exports". Please delete.*

**Author Response:**

We have removed the phrase "GEE exports" entirely from the revised manuscript as requested.
* * *
**Reviewer Comment 13:**

*L245: What is the purpose of figure 4? How does it help to understand the manuscript?*

**Author Response:**

Figure 4 from the preprint has been replaced with new figures in the revised manuscript to better support the study's objectives and improve clarity.
* * *
**Reviewer Comment 14:**

*L280: In the figure, "coefficients" is stated, but the image can only show one coefficient. I believe it is the slope (B1).*

**Author Response:**

We appreciate your attention to terminology. We used the plural term "coefficients" because the GWR produces a regression coefficient ($\beta_1$) for each pixel, reflecting spatial variation in the LST-PC1 relationship. To reduce multicollinearity and condense information from the predictors (SSM and RVI), we applied PCA to the normalized rasters and retained only the first principal component (PC1) as a composite environmental stressor. The GWR model thus regresses LST on PC1, with local coefficients varying across space. We have clarified this process in the Methodology section to avoid confusion. The figure and caption have been updated for clarity:

[Figure]

**Figure 8: Spatial distribution of Geographically Weighted Regression (GWR) model results across the Dhaka Metropolitan Area (DMA). (a) GWR coefficients for the first principal component representing spatially varying influence of Radar Vegetation Index (RVI) and Sentinel-1 Soil Moisture (SSM) on Land Surface Temperature (LST) across the DMA, with blue regions indicating areas where environmental factors provide substantial cooling effects, and red regions indicate weaker cooling relationships. (b) Spatial distribution of GWR model residuals, where blue areas indicate model overestimation of LST, red areas indicate model underestimation, and neutral-colored areas demonstrate good model-observation agreement**.
* * *
**Reviewer Comment 15:**

*L290: For GWR, you did not use observed data. Here, you make a regression using LST (from MODIS), and it is regressed using PC1.*

**Author Response:**

Correct. We have clarified in the revised manuscript that the GWR analysis uses MODIS-derived LST as the dependent variable and the first principal component (PC1) derived from SSM and RVI as the independent variable.
* * *
**Reviewer Comment 16:**

*L295: What are the units °C or Kelvin?*

**Author Response:**

All temperature values are now explicitly reported in degrees Celsius (°C) throughout the manuscript.
* * *
**Reviewer Comment 17:**

*L300: When you run a GWR you get as results spatial indicators for error (Fig. 6) and R².
You should show the spatial R² on the manuscript.*

**Author Response:**

A spatial R² map has been added in the revised manuscript (new Figure 9), showing spatial
variation in model fit across the study area.

[Figure]

**Figure 6: Spatial distribution of local R² values from the Geographically Weighted Regression (GWR) model across
the Dhaka Metropolitan Area (DMA), indicating the proportion of variance in Land Surface Temperature (LST)
explained by the first principal component (PC1) derived from Radar Vegetation Index (RVI) and Sentinel-1 Soil
Moisture (SSM) at each location. Higher values (blue) denote areas where the model provides stronger explanatory
power, while lower values (red) indicate weaker model performance.**

**Reviewer Comment 18:**
*L325: But, in L265, you say something opposite.*

**Author Response:**
We have carefully revised the relevant sentences to ensure internal consistency and avoid contradictory statements.

We would like to thank you once again for the valuable suggestions on our manuscript. We have incorporated these suggestions into the revised manuscript. We trust these revisions address your concerns and improve the clarity and rigor of the manuscript.

Aishia Fyruz Aishi

Corresponding Author

E-mail Address: aishia.fyruz@du.ac.bd